# Reconciling Registration Policies for Unmanned Aircraft with Unmanned Aircraft Ownership Characteristics

Isaac Levi Henderson 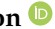

School of Aviation, Massey University, 47 Airport Drive, Palmerston North 4472, New Zealand;
i.l.henderson@massey.ac.nz; Tel.: +64-6-951-9432

**Abstract:** Registration of unmanned aircraft is a common policy around the world and forms part of the International Civil Aviation Organisation's model regulations for unmanned aircraft. This study conducts a review of the various registration policies that have been implemented amongst advanced economies to find commonalities and differences. New Zealand is then used as a case study. The country does not currently have registration of unmanned aircraft; however, their Ministry of Transport has put forward the idea of implementing a registration scheme. As part of this case study, the ownership characteristics of 919 New Zealand unmanned aircraft users were collected using an online survey. The results highlight that personally owned aircraft tend to only be used by their owner, with the number of users being lower than the number of aircraft. For organisationally owned aircraft, there are multiple users per aircraft; however, these users tend to only be employees of the organisation. These findings suggest that for New Zealand, the best way to implement a registration scheme would be to register users and organisations rather than individual aircraft. While specific to New Zealand, these findings also prompt the need for future research worldwide to see whether registration schemes reconcile with ownership data.

**Keywords:** drone registration; unmanned aircraft; transport policy

## 1. Introduction

Registration for unmanned aircraft (also known as drones, model aircraft, uncrewed aircraft, and remotely-piloted aircraft) has been a common policy around the world, with the United States, European Union countries, and many other jurisdictions implementing such schemes within the last decade. While a registration scheme for unmanned aircraft has been proposed in New Zealand within a discussion document from the Ministry of Transport [1], it is not official government policy, and the discussion document prompted feedback from stakeholders as to whether they support such a scheme and what that might look like. While a more detailed comparison is provided in Table 1, essentially all advanced economies have registration schemes, but they are divided as to whether it is the unmanned aircraft itself that should be registered, or whether the user should be registered and assigned an operator ID that can be affixed to multiple aircraft. New Zealand could choose to adopt either system, or to not implement a registration scheme at all. In this sense, ownership data observed from users can help inform future government policy.

Different benefits and reasoning have been provided by nations when proposing registration schemes. Some of the key reasons include:

- Creating a system that encourages responsibility and accountability
- Enabling an unmanned aircraft user to recover their lost aircraft
- Promoting public acceptance of unmanned aircraft technologies by providing mechanisms to ensure that owners can be identified
- Creating a safer and more secure aviation system
- Better data provided to government agencies to help inform future policies
- Ability to directly communicate with unmanned aircraft users

- Ability to identify non-compliant operators and proceed with enforcement action where required
- Facilitating airspace integration by allowing unmanned aircraft and manned aircraft to operate in the same airspace
- Allowing for a wider range of operations to take place with unmanned aircraft [1–7].

The academic literature provides differing perspectives on the topic of unmanned aircraft registration. Clarke [8] argues the need for registration using the logic that this is what is currently required for road vehicles, though suggests that the systems that were being implemented in the United States at the time would provide little beneficial effect and instead should have been more onerous to push users to "join a club and bring themselves within an environment that makes information available, and brings with it both a sense of responsibility and insurance coverage" (pp. 154–155). Merkert et al. [9] find that the willingness of Australian unmanned aircraft users to pay for aircraft registration depends on the types of operations that they are able to conduct with that registration. They suggest that policymakers should not tax the growth of unmanned aircraft use through annual registration charges because the tokenistic charges are below what users are willing to pay, and because they will not be popular or effective as users will not see what they get in return for the charges. Yao et al. [10] find that owner registration received the most support from both unmanned aircraft pilots and bystanders as a potential mechanism to alleviate privacy concerns using a sample from the United States. The reasons for this support were because it is practical in tracking down pilots, there are similar mechanisms in other domains, and it discourages irresponsible use (the latter only perceived among bystanders). However, both pilots and bystanders raised the issue that it does not directly protect privacy, and pilots were concerned that owner registration could breach their own rights to privacy.

In 2019, Shelley [11] concluded that given current technologies, the Gatwick incident could not have been prevented by a registration scheme because those who want to conduct malicious and intentional attacks will circumvent any existing registration or remote identification requirements. Conversely, Pentz and Tang [12] do find that the registration of unmanned aircraft in the United States did have an effect on reducing unauthorised and careless events using a case study around 2017 where the courts stopped the registration scheme, and then it was re-initiated. Likewise, Pitcher [13] found that the number of unmanned aircraft sightings submitted to the Federal Aviation Administration (FAA) increased significantly in 2018 but did not increase in 2019. Given the reintroduction of a registration requirement in December 2017, they argue that this stagnation in sightings in 2019 was likely due to the effect of the registration scheme as no other changes happened at a federal level during this time period. There have also been other papers that recommend registration schemes [14–18], though these are not supported with empirical data and are instead argumentations drawn from various perspectives (e.g., legal, societal, law enforcement). Given the limited and mixed evidence and opinions within the literature as to whether registration schemes are effective, this paper will not wade into this debate. Rather, the ownership data of New Zealand unmanned aircraft users can be used to inform what sort of registration scheme should be implemented in New Zealand should it decide to implement one. New Zealand also has the benefit of being able to be informed by systems of other jurisdictions, which this paper will also help to elucidate.

### 1.1. An International Review of Registration Schemes for Unmanned Aircraft

New Zealand shares many similarities with other jurisdictions with regard to the regulation of unmanned aircraft, but is unusual in not having any form of registration system for unmanned aircraft [19,20]. However, internationally, there are still significant differences in what registration systems are used, particularly as to whether each aircraft is registered individually, or whether each user (usually referred to as an operator) is registered and has a unique code that they can affix to their unmanned aircraft. Table 1 presents a summary of differences between the registration policies for unmanned aircraft across

advanced economies. The list of advanced economies was taken from the International Monetary Fund [21]. Only those jurisdictions where details from an official government website could be accessed without starting an application were included.

**Table 1.** Comparison of unmanned aircraft registration schemes across advanced economies.

| Jurisdiction | Registration Scheme | Threshold | Cost | Duration | Age Requirement | Exceptions |
|---|---|---|---|---|---|---|
| Australia | Every aircraft used for commercial purposes must be registered | None for commercial 250 g for recreational (mid-2023) | AUD 40 (USD 26.72) per aircraft | 1 year | 16 years | Recreational use (until mid-2023) |
| Austria [1] | Each aircraft must be marked with the Operator ID for the owner | 250 g or equipped with camera | EUR 32.40 (USD 34.43) per operator | 3 years | 18 years | - |
| Canada | Every aircraft must be registered | 250 g | CAD 5 (USD 3.68) per aircraft | Not specified | 14 years | - |
| EASA Member States [2] | Each aircraft must be marked with the Operator ID for the owner | 250 g, equipped with sensor that can collect personal data, or has potential human kinetic energy of more than 80 joules | Varies | Varies | Varies | Nations can adopt different rules for model aircraft associations |
| Finland [1] | Each aircraft must be marked with the Operator ID for the owner | 250 g or equipped with camera | EUR 30 for 1 year, EUR 75 for 3 years, or EUR 100 for 5 years per operator (USD 31.87, USD 79.67, or USD 106.23) | Not specified | - |
| France [1] | Each aircraft must be marked with the Operator ID for the owner | EASA Requirements | Not specified | Not specified | Not specified | Different rules are applied for members of model aircraft associations, though registration is usually still required |
| Germany [1] | Each aircraft must be marked with the Operator ID for the owner | EASA Requirements | Not specified | Not specified | Not specified | - |
| Hong Kong (SAR) | Every aircraft must be registered | 250 g | Free (until 1 June 2025) | Up to 5 years | 18 years | - |
| Iceland [1] | Every aircraft used for commercial purposes must be registered | 250 g | Free | Not specified | Not specified | Recreational use |

**Table 1.** *Cont.*

| Jurisdiction | Registration Scheme | Threshold | Cost | Duration | Age Requirement | Exceptions |
|---|---|---|---|---|---|---|
| ICAO Model Regulations | Every aircraft must be registered | Not specified | Not specified | Not specified | Not specified | Different provisions for approved aviation organisations |
| Ireland [1] | Each aircraft must be marked with the Operator ID for the owner | 250 g or equipped with camera | EUR 30 (USD 31.88) per operator | 2 years | 16 years | - |
| Japan | Every aircraft must be registered | 100 g | Online application costs ¥900 for first aircraft and ¥890 for subsequent aircraft (USD 6.84/6.76) | 3 years | Not specified | - |
| Luxembourg [1] | Each aircraft must be marked with the Operator ID for the owner | 250 g, equipped with camera, or can operate above 90 km/h | Not specified | 1 year | Not specified | - |
| Macau (SAR) | None—but all drone activities must be approved | - | - | - | - | - |
| Netherlands [1] | Each aircraft must be marked with the Operator ID for the owner | 250 g or equipped with camera | EUR 23 (USD 24.44) per operator | 1 year | 16 years | - |
| New Zealand | None—but Part 102 operators must list aircraft and unique identifiers in exposition | - | - | - | - | - |
| Singapore [1] | Every aircraft must be registered | 250 g | SGD 15 (USD 11.10) per registration label | Not specified | 16 years [3] | - |
| Slovenia [1] | Each aircraft must be marked with the Operator ID for the owner | EASA Requirements | Not specified | Not specified | 18 years [3] | - |
| South Korea | Every aircraft must be registered | 2 kg for recreational use, all commercial use | Not specified | Not specified | Not specified | - |

**Table 1.** *Cont.*

| Jurisdiction | Registration Scheme | Threshold | Cost | Duration | Age Requirement | Exceptions |
|---|---|---|---|---|---|---|
| Sweden [1] | Each aircraft must be marked with the Operator ID for the owner | Not specified | 130 SEK (USD 12.46), covering license and operator ID | 5 years | Not specified | - |
| Switzerland [1] | Each aircraft must be marked with the Operator ID for the owner | 250 g or equipped with a sensor that can record personal data | Not specified | Not specified | 12 years | - |
| Taiwan [1] | Every aircraft must be registered | 250 g | TWD 50 (USD 1.63) | Not specified | 20 years [3] | - |
| United Kingdom [1] | Each aircraft must be marked with the Operator ID for the owner | 250 g or equipped with camera | £10 (USD 12.18) per person or organisation | 1 year | 18 years [3] | - |
| United States | Every aircraft must be registered | 250 g | USD 5 per aircraft | 3 years | 13 years | Exception for recreational flyers, who must follow the rules of an FAA-recognised Community-Based Organisation |

[1] Has other requirements that must be met when applying for registration, such as pilot competency, theory tests, training, airworthiness requirements, insurance requirements, or a limiting the number of aircraft that can be registered. [2] EASA Member States include the 27 European Union Countries, as well as Iceland, Norway, Liechtenstein, and Switzerland. EASA Members are also included in the table separately where information was accessible. [3] Younger persons can apply through their parents, guardians, or legal representatives. Table assumes that the aircraft is below 25 kg and will be operated outdoors. It simplifies requirements to allow for easy comparison on registration requirements. Readers should consult the relevant jurisdiction's rules for full details and to check for updates. Currency conversions were done using rates for 21 December 2022 [2,4,5,22–47].

As can be seen from Table 1, only New Zealand and the Special Administrative Region of Macao do not require registration for unmanned aircraft or users. While substantial jurisdictions like the United States, Canada, Australia, Japan, South Korea, Singapore, and Taiwan have adopted aircraft registration schemes, member nations of the European Union Aviation Safety Agency (EASA) as well as the United Kingdom have adopted registration for operator IDs, which can be applied to multiple aircraft. Most of these latter group of nations also have training, competency, and insurance requirements to accompany operator registration.

*1.2. New Zealand's Proposed Registration Scheme*

In April 2021, the New Zealand Ministry of Transport released the *Enabling Drone Integration* discussion document, which proposed a number of changes to the regulatory framework for unmanned aircraft in New Zealand. This included a proposal for registration of unmanned aircraft. The discussion document highlights that the problems that registration is aiming to solve are (quoting verbatim):

- Lack of compliance from drone pilots
- Inability to communicate directly with drone owners

- Inability to link a drone to a person, i.e., inability to identify who flies what, making complaints often unenforceable
- Limited ability to integrate drones into the civil aviation system.
- Difficulty in collecting accurate data and managing risks accordingly (p.41, [1]).

The proposal suggests that in most instances the owner of the unmanned aircraft would be the one who is required to register an aircraft, though they expect that the pilot and the owner will usually be the same. Owners may also be organisations such as companies, government departments, and incorporated societies. The weight threshold of 250 g was proposed for requiring registration, wanting to remain in line with the "majority of our aviation counterparts" (p.42, [1]) and because this threshold has been supported by research [48]. The age requirement proposed was 14 years, with younger persons able to register an unmanned aircraft through their parent or guardian. This is lower than most jurisdictions in Table 1, but the logic is unique to New Zealand: 14 is the youngest that a person can be issued an infringement notice and the minimum age for authentication of identity services. Analogous to the United States' exceptions for community-based organisations, the discussion document proposes that unmanned aircraft operated within Model Flying New Zealand (MFNZ) designated areas and under supervision of MFNZ would not be required to be registered. However, any operations outside of MFNZ designated areas would require registration, which would still limit operations for many MFNZ members who may fly over their private property or at public parks. Unlike the other jurisdictions in Table 1, the discussion document proposes that registration should not expire and be one-off for each aircraft. No estimate of cost was provided for registration, though the document does outline that "any potential new fee should not become a barrier or disincentive to comply" (p.48, [1]). The discussion document also says that "we would need to consider whether to charge either per drone or per owner and whether this should be a one-off or annual fee" (p.48, [1]) As it is a discussion document that seeks public submissions, and not government policy, it is unclear as to which system will eventually be used (or even if a registration scheme will be implemented). In particular, New Zealand can opt to register each aircraft, or to register each operator—allowing each operator to affix their unique identifier to multiple aircraft. Both approaches are being used by major jurisdictions, so in this sense New Zealand should be guided by evidence from its own unmanned aircraft user population, particularly around unmanned aircraft ownership. This study aims to provide an informative contribution to help guide the eventual policy direction that New Zealand takes.

*1.3. New Zealand's Current Unmanned Aircraft Regulations*

Unmanned aircraft operations are regulated under Part 101 and Part 102 of the Civil Aviation Rules (CARs), which were created in accordance with the Civil Aviation Act [49,50]. CAR Part 101 can be thought of as a set of general operating rules, which anyone can operate under provided that they stay within its bounds. Some of the bounds include:

1. Operations must take place during daylight, unless indoors or within a shielded area (below the height of the tallest object within 100 m of the aircraft).
2. Operations must be conducted within visual line of sight, which may include the use of trained and competent visual observers.
3. Flights must not exceed 400 ft above ground level, unless a Notice to Airmen has been issued at least 24 h prior to the operation.
4. Operations must not be flown over uninvolved persons, unless their consent has been obtained.
5. Operations may only be flown over property with the consent of the owner or occupier of that property.
6. The same airspace restrictions apply as for manned aircraft, such as the need to obtain permission from air traffic control to enter controlled airspace, and the requirement to comply with the requirements of special use airspace. The only exception is Low Fly Zones, which cannot be entered under CAR Part 101.

7. Operations can only take place within 4 km of a promulgated aerodrome if the operation is shielded and there is a physical barrier between the unmanned aircraft and the aerodrome; or, if the pilot holds an approved qualification or is under the direct supervision of someone who does hold such a qualification.

Aircraft above 15 kg, but less than 25 kg may be operated, but must be inspected and approved by a person or organisation authorised by the Civil Aviation Authority. Operations of aircraft above 25 kg are not permitted under CAR Part 101 [50].

Any operations outside of the bounds outlined above must be conducted under the authority of a Part 102 Operator's Certificate. Organisations may apply for such a certificate by submitting an exposition for consideration [37,49]. This document should outline how the organisation is managed, standard operating procedures, maintenance and airworthiness procedures, training requirements, and safety systems appropriate for the size and complexity of the operation [51].

## 2. Materials and Methods

This study forms part of a wider research project examining unmanned aircraft users within New Zealand. Part of this research project was the running of an online survey that was hosted via Qualtrics and ran from October 2020 until January 2021. This survey asked about the ways unmanned aircraft were being used, levels of aircraft ownership, effectiveness of safety promotion, the fitness of the regulatory framework, and other topics of interest. Participants were recruited by posting the recruitment link onto social media forums, and by asking Model Flying New Zealand, UAVNZ, and Aviation New Zealand to share the link with their members. While it was open, the survey yielded 946 responses. However, only 919 responses were complete enough to be useful for the study.

To ensure a useful sample, users had to meet the following recruitment criteria:

1. Reside in New Zealand
2. Have flown an unmanned aircraft before
3. Be 16 years or older

The study was peer-reviewed and deemed to be low-risk in terms of potential ethical issues. Thus, it was not sent to one of Massey University's Human Ethics Committees for approval, but was registered as a low-risk study in Massey University's Human Ethics Database.

This study only analyses a subset of the total questions asked in the wider survey. Appendix A provides a list of questions that are analysed in this study. The full dataset for the survey (including all questions and responses) has been made publicly available [52] and can be accessed here: https://doi.org/10.6084/m9.figshare.16571558.v1 (accessed on 13 January 2023).

All 919 valid responses were included for this study as all these participants answered the questions related to aircraft ownership (Q4–12—see Appendix A); however, 28 participants did not answer the questions about having attended a course, completing an OCA, being members of MFNZ or UAVNZ, or having operated under a Part 102 Operator's Certificate before (Q13–17).

Of the 919 participants, 897 (97.61%) identified as male, 17 (1.85%) as female, 1 (0.11%) as transgender, and 4 (0.44%) preferred not to say their gender. The mean age of the sample was 57.05 (SD = 15.76), with the youngest participant being 16 years old and the eldest 88 years old. There were 38 (4.13%) participants that were non-current users of unmanned aircraft, 777 (84.55%) who were recreational users, 59 (6.42%) who were semi-professional users (less than 50% of work time spent on unmanned aircraft-related activities), and 45 (4.90%) who were professional users (more than 50% of work time spent on unmanned aircraft-related activities). Out of the 891 participants who answered questions 13–17 (see Appendix A), 416 (46.69%) had attended a course on unmanned aircraft operations, 617 (69.25%) had passed an OCA, 707 (79.35%) were members of MFNZ, and 123 (13.80%) were members of UAVNZ or Aviation New Zealand.

Because the different questions in Appendix A yield different types of data, different statistical analyses were performed based upon the question type. Questions 1 and 2 were not further analysed as neither gender nor age are relevant considerations for aircraft ownership. Questions 3, and 13–17 were used to categorise participants into groups based upon different user characteristics. Questions 4, 5, 8, and 9 are all yes/no questions related to aircraft ownership. To examine whether there are statistically significant associations between a participant answering yes and being from a particular user grouping, Chi-squared tests of independence were used, with effect size reported as Cramer's V [53]. Questions 6, 7, and 10–12 provide numerical values as answers. However, these are not normally distributed. Accordingly, non-parametric tests are used. Kruskal–Wallis H tests [54] were used to see whether differences in the number of aircraft/users from each of these questions were significantly different based on user type (non-current, recreational, semi-professional, or professional). Distributions were checked for similarity by visual inspection of a box plot. Pairwise comparisons were performed using Dunn's procedure [55] with a Bonferroni correction [56] applied for multiple comparisons. Mann–Whitney U tests [57] were performed to see whether differences in the number of aircraft/users from each of these questions were significantly different based upon having attended a course, having passed an OCA, being a member of MFNZ or UAVNZ, or having operated under a Part 102 Operator's Certificate before. Distributions were assessed to be similar based upon visual inspection. Results are reported according to mean ranks and distributions using an exact sampling distribution for *U* [58].

## 3. Results

### 3.1. Descriptive Statistics on Ownership and Number of Users for Unmanned Aircraft

Table 2 collates the raw number and percentages of users that have personally owned aircraft either now or in the past, as well as those whose organisations own aircraft either now or in the past.

**Table 2.** Ownership rates for unmanned aircraft.

| Ownership Variable | Participants | |
|---|---|---|
| | **Number** | **Percentage** |
| Personally owned an unmanned aircraft in the past | 879 | 95.65% |
| Currently owns an unmanned aircraft [1] | 857 | 93.25% |
| Organisation has owned an unmanned aircraft in the past [2] | 278 | 30.25% |
| Organisation currently owns an unmanned aircraft [2] | 222 | 24.16% |

[1] Excludes 40 participants who indicated they had not owned an unmanned aircraft in the past. [2] Excludes 237 participants who answered "unsure" to organisational ownership as well as the 5 participants who did not answer these questions.

Table 3 presents descriptive statistics in terms of the number of presently owned aircraft and other users of those aircraft in the last 12 months. There was positive skewness present for each variable, hence, while mean and standard deviation are reported, the more useful variables are arguably the median and inter-quartile range, shown in the "Median (IQR)" column. The number outside brackets is the median (i.e., the middle of the observed values), the first number in the brackets is the lower quartile (i.e., the 25th percentile), and the second number is the upper quartile (i.e., the 75th percentile). The range is also provided to give an idea of the extremes of the distribution.

**Table 3.** Descriptive statistics regarding number of personally and organisationally owned aircraft as well as the number of users of those in the last 12 months.

| Ownership Variable | N ^ | Mean (SD) | Median (IQR) | Range |
|---|---|---|---|---|
| Personally owned aircraft per user | 919 | 11.26 (14.31) | 8 (3,15) | 0–250 |
| Other users of personally owned aircraft | 856 | 1.08 (2.40) | 0 (0,1) | 0–30 |
| Total users of personally owned aircraft | 856 | 2.08 (2.40) | 1 (1,2) | 1–31 |
| Users per personally owned aircraft | 856 | 0.43 (0.79) | 0.2 (0.1,0.5) | 0.01–11 |
| Organisationally owned aircraft | 218 | 8.83 (21.35) | 2 (1,6) | 1–200 |
| Employee users of organisational aircraft | 214 | 6.48 (23.68) | 2 (1,4) | 0–275 |
| Non-employee users of organisational aircraft | 214 | 2.24 (8.25) | 0 (0,1) | 0–97 |
| Total users of organisational aircraft | 214 | 8.73 (24.66) | 3 (1, 6) | 0–275 |
| Users per organisationally owned aircraft | 214 | 2.33 (6.36) | 1 (0.41, 2) | 0–82.5 |

^ The number of participants varies because some participants did not answer follow-up questions or were excluded due to earlier answers.

### 3.2. Differences between Users

Table 4 presents the statistically significant results from the Chi-squared tests of independence (to see if particular user groups are more or less associated with owning an aircraft personally or organisationally), Kruskal–Wallis H tests (to examine whether user type affects the number of owned unmanned aircraft and number of users for those aircraft), and Mann–Whitney U tests (to examine whether having attended a course, passed an OCA, being a member of MFNZ or UAVNZ, or having operated under a Part 102 Operator's Certificate affects the number of owned unmanned aircraft and number of users for those aircraft). Full statistical reporting, including statistically insignificant results are reported in Appendix B. Table 3 presents the user grouping where statistically significant differences were observed and indicates the directionality of this effect with either a (+) sign to indicate greater likelihood or a higher number, or a (-) symbol to indicate lesser likelihood or a lower number when compared with the rest of the sample. In the case of pairwise comparisons, these are always presented in terms of which group had a higher mean rank than the other using the greater than symbol (>).

**Table 4.** Statistically significant differences between user groupings for ownership variables.

| Ownership Variable | N ^ | Observed Differences |
|---|---|---|
| User personally owns an aircraft | 919 | Recreational (+) *** <br> Non-Current User (-) *** |
| | 891 | Passed an OCA (+) *** <br> MFNZ Member (+) *** <br> Part 102 Operator (-) * |
| Number of personally owned aircraft | 889 | Recreational User > Non-Current User *** <br> Recreational User > Semi-Professional User *** <br> Recreational User > Professional User *** <br> Completed a Course (+) ** <br> Passed an OCA (+) *** <br> MFNZ Member (+) *** |
| Other users of personally owned aircraft | 856 | Recreational User > Non-Current User * <br> Professional User > Non-Current User * |

**Table 4.** *Cont.*

| Ownership Variable | N ^ | Observed Differences |
|---|---|---|
| Total users of personally owned aircraft | 856 | Recreational User > Non-Current User * <br> Professional User > Non-Current User * |
| Users per personally owned aircraft | 856 | Recreational User > Semi-Professional User *** |
| | 828 | Recreational User > Professional User *** <br> Completed a Course (-) * <br> Passed an OCA (-) *** <br> MFNZ Member (-) *** |
| User's organisation owns an aircraft | 676 | Semi-Professional User (+) *** |
| | 659 | Professional User (+) *** <br> Completed a Course (+) *** <br> Passed an OCA (+) *** <br> MFNZ Member (-) *** <br> UAVNZ Member (+) *** <br> Part 102 Operator (+) *** |
| Number of organisationally owned aircraft | 218 | Professional > Recreational ** |
| | 212 | Completed a Course (+) *** <br> Part 102 Operator (+) ** |
| Employee users of organisationally owned aircraft | 212 | UAVNZ Member (+) * |
| Non-employee users of organisationally owned aircraft | 214 | User Type (pairwise comparisons insignificant) * |
| | 212 | Part 102 Operator (-) ** |
| Total users of organisationally owned aircraft | 212 | UAVNZ Member (+) * <br> Part 102 Operator (-) * |
| Users per organisationally owned aircraft | 214 | Recreational > Professional *** |
| | 212 | Completed a Course (-) *** <br> Passed an OCA (-) * <br> Part 102 Operator (-) *** |

^ The number of participants varies because some participants did not answer follow-up questions or were excluded due to earlier answers. *, **, *** indicate statistical significance at the $p < 0.05$, 0.01, and 0.001 levels, respectively.

## 4. Discussion

The results reveal two different pictures based upon whether an unmanned aircraft is personally owned or organisationally owned. Personally owned aircraft only had a mean of 0.43 users per aircraft, or a median of 0.2. In this sense there are more aircraft than there are people using them. The opposite is true for organisationally owned aircraft, with a mean of 2.33 users per aircraft, or a median of 1. In the majority of cases, all users of organisational aircraft were employees of the organisation; however, this was not always the case with one participant reporting as many as 97 non-employee users of their organisations' aircraft. Even as descriptive statistics, these results suggest that New Zealand should adopt an approach similar to EASA member nations and the United Kingdom, whereby operators are registered rather than aircraft. Operators can be individuals or organisations, with their unique operator IDs needing to be affixed to each aircraft. Due to the limitations of this study, it is unclear whether non-employee users of organisationally owned aircraft were being supervised by an employee when using the aircraft. However, one solution would be to separate flyer ID (i.e., the person who is flying the aircraft) from operator ID (i.e., the individual or organisation who owns the aircraft), as is done in the United Kingdom. This would also align with one of the other proposals in the *Enabling Drone Integration* discussion document, which was to have a basic pilot qualification to be able to pilot unmanned aircraft above 250 g.

Out of the statistically significant differences observed between users in Table 4, there are a few that stand out as important for informing policy. Firstly, it is recreational users who own the most aircraft, more than all other user types. MFNZ Members also own more aircraft than non-members. This suggests that an exception being made for MFNZ members (as is proposed in the discussion document) would be beneficial for MFNZ members by avoiding unnecessary compliance costs for flying model aircraft within designated areas and under MFNZ rules. However, this exception would only be important if New Zealand followed through with requiring each aircraft to be registered rather than each operator. This study observed one participant (who was an MFNZ member) who personally owned 250 aircraft. Taking a cost of NZD 10 (USD 6.29) as an estimate per aircraft per annum (based upon charges observed internationally), that would cost that user NZD 2500 (USD 1573.74) every year to be able to fly their full set of aircraft outside of MFNZ designated areas. However, if the charge was per operator and they were able to affix their operator ID to all 250 of their aircraft, then the costs would be far more tolerable. The United Kingdom and most EASA member countries require operator registration and operator IDs to be fixed to model aircraft as with any other unmanned aircraft. MFNZ members were also less likely to let other users fly their aircraft, despite recreational users generally sharing their aircraft amongst users more than semi-professional or professional users. This again suggests that either the exception or adoption of an operator ID type of system would be more equitable for MFNZ members.

Unsurprisingly, organisational ownership of unmanned aircraft was higher amongst semi-professional and professional users. Organisational ownership was also more common amongst those who had completed a course, those who had passed an OCA, those who were members of UAVNZ, and those who operate under Part 102 Operator's Certificates. Past research has shown that all of these variables improve the number and variety of operational risk mitigations applied by users [59]. Organisations are potentially more risk averse because of the potential liabilities (both financial and in terms of public perceptions) if something wrong were to happen with one of their aircraft. Importantly, they can regulate requirements for employees to be able to operate aircraft, and in what situations non-employees may be allowed to pilot an organisationally owned aircraft. In this sense, registration of the organisation with an operator ID (rather than each aircraft having its own unique ID) makes more sense as this is where the accountability lies. The same issues are also present as for MFNZ members, with high ownership rates amongst some organisations. One participant reported that their organisation owned 200 aircraft. Even if there was a higher charge for an organisation to register an operator ID, it would still be cheaper than an annual aircraft registration bill that would likely be around NZD 2000 (USD 1258.99). It would also help alleviate the inefficiencies associated with having to register each individual aircraft as they are purchased given that organisations may be using multiple models and upgrading them regularly. A unique operator ID would be able to be affixed to all organisationally owned aircraft. It is also worth highlighting that organisations certificated under Part 102 will have already paid significant fees to be able to achieve that certification from the Civil Aviation Authority, and will already have unique identifiers for each of their aircraft. This study shows that they have lower numbers of users per aircraft, and larger numbers of aircraft overall, suggesting that a transition to an operator ID system would help prevent additional costs upon this user group.

## 5. Conclusions

This study analysed unmanned aircraft ownership characteristics among a sample of 919 New Zealand unmanned aircraft users. Understanding these characteristics can inform future policies regarding registration of unmanned aircraft. A global review of policies in advanced economies reveals that the two most common approaches are registering each unmanned aircraft individually, or registering each operator and assigning them an operator ID that must be displayed on each aircraft that they own/operate. The results of this study suggest that the latter system of operator IDs (used by EASA member nations

and the United Kingdom) would be the best option for New Zealand, whereby operators (individuals or organisations) and their unique operator IDs are displayed on all aircraft that they fly. Differences among users suggest that if New Zealand were to require individual aircraft registration, it would be particularly costly for MFNZ members who own greater numbers of aircraft, and for organisations that use unmanned aircraft professionally due to their higher ownership rates. While the Ministry of Transport has proposed an exception for MFNZ members within their discussion document that proposes individual aircraft registration for aircraft over 250 g, this would only apply to aircraft flown at MFNZ sites, which would significantly limit their current freedoms. Therefore, this paper suggests that the best registration policy for New Zealand would be to require all unmanned aircraft users to register for an operator ID (usually the owner, except in exceptional circumstances) and display it on all aircraft they fly. MFNZ members could receive the same exception currently proposed for flights at MFNZ sites, but would still be subject to registration requirements when operating elsewhere. An operator ID system is unlikely to impose significant financial constraints on MFNZ members or organisations like an aircraft-based system would.

The effectiveness of registration schemes is unclear, as many claims about their benefits are not yet supported by empirical evidence. In this sense, New Zealand may choose to continue to maintain its current policy of having no registration scheme until there is more established evidence. The decision on whether to implement a registration scheme will be made by New Zealand's Cabinet, who as elected officials will need to consider the public interest. While the evidence presented in this study cannot directly inform whether a registration scheme should be implemented, it does provide insight into what such a scheme should look like if implemented.

## 6. Limitations and Future Research

This paper's key limitation is that the data obtained are only from New Zealand users. While this is very helpful for informing policy decisions in New Zealand, the data may not generalise to other jurisdictions. However, it is clear that simply obtaining ownership information from individual and organisational users can help to inform what might be a fairer and more effective registration scheme within a jurisdiction. In this sense, further studies in other jurisdictions would help better inform discussion and debate. For developing nations, which may have less administrative capability, this information can also help to see what might be the most straightforward was of implementing such a system and whether that is a worthwhile exercise or not given the need to prioritise tasks with limited resources. Similarly, there seems to be very little data in the literature to support registration schemes as having an effect upon safety, or for providing other benefits. This gap in the literature is startling considering the extent to which schemes have been implemented by governments around the world. If the benefits cannot be measured or substantiated, then this raises issues of legitimacy and proportionality. The inability to communicate clear benefits to users has been raised as an issue in the literature [9], yet there seems to be little attempt to show the benefits of these systems once they have been implemented. This is an obvious area for future research to tackle, though may be limited by government agencies' willingness to share data.

**Funding:** This research received no external funding.

**Data Availability Statement:** The data that underlie this article are publicly available at the following link: https://doi.org/10.6084/m9.figshare.16571558.v1 (accessed on 13 January 2023).

**Conflicts of Interest:** Isaac Henderson is the Chair of UAVNZ, and industry and professional body representing commercial unmanned aircraft operators and other organisations that play a role in commercial unmanned aerospace. Isaac Henderson also provides consultancy services for unmanned aircraft operators, particularly with regard to helping achieve certification under Part 102 of the Civil Aviation Rules. However, none of the findings of this research benefit him directly or indirectly.

**Appendix A. Extracted Survey Questions**

This Appendix provides a list of questions that were extracted from the wider survey data. The question numbering is ordered according to the order the participants saw each question, however, does not match that of the wider survey (which has additional questions in-between and after those listed below).

1.  What is your gender?
    a.   Male
    b.   Female
    c.   Other (please specify)
    d.   Prefer not to say
2.  What is your age?
3.  Which of the following best describes you?
    a.   Not a current unmanned aircraft user
    b.   Recreational unmanned aircraft user
    c.   Semi-professional commercial unmanned aircraft user (i.e., where less than 50% of your work time is spent on activities related to unmanned aircraft, including flight time, travel time, maintenance, data processing, etc.)
    d.   Professional commercial unmanned aircraft user (i.e., where more than 50% of your work time is spent on activities related to unmanned aircraft, including flight time, travel time, maintenance, data processing, etc.)
4.  Have you ever personally owned an unmanned aircraft?
    a.   Yes
    b.   No
5.  [If answer to Q7 is yes] Do you currently personally own an unmanned aircraft?
    a.   Yes
    b.   No
6.  [If answer to Q8 is yes] How many unmanned aircraft do you personally own?
7.  [If answer to Q8 is yes] How many other people have operated one of the unmanned aircraft that you personally own within the last 12 months?
8.  Has your organisation ever owned an unmanned aircraft?

    a.   Yes
    b.   No
    c.   Unsure
9.  [If answer to Q11 is yes] Does your organisation currently own any unmanned aircraft?
    a.   Yes
    b.   No
10. [If answer to Q12 is yes] How many unmanned aircraft does your organisation own?
11. [If answer to Q12 is yes] How many people (including yourself) have operated one of your organisation's unmanned aircraft within the last 12 months?
12. [If answer to Q12 is yes] Out of those people who have operated one of your organisation's unmanned aircraft within the last 12 months, how many were employees or otherwise involved with the organisation?
13. Have you ever done a course on unmanned aircraft operations?
    a.   Yes
    b.   No
14. Have you ever passed an operational competency assessment (also known as a flight examination) on an unmanned aircraft?
    a.   Yes
    b.   No

15. Are you a member of Model Flying New Zealand?
    a. Yes
    b. No

16. Are you or your organisation a member of UAVNZ and/or Aviation New Zealand?
    a. Yes
    b. No

17. Have you ever operated under a Part 102 Operator's Certificate?
    a. Yes
    b. No

**Appendix B. Full Statistical Reporting**

This appendix provides full statistical reporting for the Chi-squared tests of independence, Kruskal–Wallis H tests, and Mann–Whitney U tests. It includes statistically insignificant results, which were not reported in the body of the manuscript.

Chi-squared tests of independence yielded the following results (where significant, effect size is reported with Cramer's V, where 0.1, 0.3, and 0.5 indicate small, medium, and large effect sizes, respectively [53]):

1. Recreational users were more likely to personally own an unmanned aircraft, and non-current users less likely compared with the rest of the sample, $\chi^2(3) = 238.353$, $p < 0.001$, V = 0.509.
2. No statistically significant differences were observed between personally owning unmanned aircraft and having attended a course, $\chi^2(1) = 0.264$, $p = 0.352$.
3. Those who had passed an OCA were more likely to personally own an unmanned aircraft, $\chi^2(1) = 13.618$, $p < 0.001$, V = 0.124.
4. MFNZ members were more likely to personally own an unmanned aircraft, $\chi^2(1) = 27.752$, $p < 0.001$, V = 0.176.
5. No statistically significant differences were observed between being a UAVNZ member and personally owning an unmanned aircraft, $\chi^2(1) = 2.873$, $p = 0.071$.
6. Those who had operated under a Part 102 Operator's Certificate were less likely to personally own an unmanned aircraft, $\chi^2(1) = 6.555$, $p = 0.014$, V = 0.086.
7. Semi-professional and professional users were more likely to have their organisation own unmanned aircraft, $\chi^2(3) = 164.317$, $p < 0.001$, V = 0.493.
8. Those who had completed a course were more likely to have their organisation own unmanned aircraft, $\chi^2(1) = 39.792$, $p < 0.001$, V = 0.246.
9. Those who had passed an OCA were more likely to have their organisation own unmanned aircraft, $\chi^2(1) = 12.900$, $p < 0.001$, V = 0.140.
10. MFNZ members were less likely to have their organisation own unmanned aircraft, $\chi^2(1) = 18.932$, $p < 0.001$, V = 0.169.
11. UAVNZ members were more likely to have their organisation own unmanned aircraft, $\chi^2(1) = 12.705$, $p < 0.001$, V = 0.139.
12. Users who had operated under a Part 102 Operator's Certificate were more likely to have their organisation own unmanned aircraft, $\chi^2(1) = 28.613$, $p < 0.001$, V = 0.208.

Kruskal–Wallis H tests yielded the following results (where significant pairwise comparisons according to Dunn's procedure [55] are also reported):

1. There were statistically significant differences between the number of personally owned aircraft based upon user type, $\chi^2(3) = 142.004$, $p < 0.001$. Pairwise comparisons revealed significant differences in the mean rank number of personally owned aircraft between recreational users (502.67) and non-current users (143.47, $p < 0.001$), semi-professional users (256.12, $p < 0.001$), and professional users (231.30, $p < 0.001$). No other pairwise comparisons were statistically significant.

2. There were statistically significant differences between the number of other users for personally owned aircraft based upon user type, $\chi^2(3) = 9.302$, $p = 0.026$. Pairwise comparisons revealed significant differences in the mean rank number of other users for personally owned aircraft between non-current users (274.70) and both recreational users (429.80, $p = 0.044$) and professional users (483.98, $p = 0.016$). No other pairwise comparisons were statistically significant.

3. The same results as in point 2 were observed for the total number of users for personally owned aircraft (this is because total users is just number of other users plus 1 for the owner of the aircraft).

4. There were statistically significant differences between the number of users per personally owned unmanned aircraft based upon user type, $\chi^2(3) = 58.212$, $p < 0.001$. Pairwise comparisons revealed significant differences in the mean rank number of users per personally owned aircraft between recreational users (405.80) and both semi-professional users (610.86, $p < 0.001$) and professional users (632.95, $p < 0.001$). No other pairwise comparisons were statistically significant.

5. There were statistically significant differences between the number of organisationally owned unmanned aircraft based upon user type, $\chi^2(3) = 16.601$, $p < 0.001$. The only statistically significant pairwise comparison showed that the mean rank number of organisationally owned aircraft was higher for professional users (mean rank = 140.88) than for recreational users (mean rank = 100.24, $p = 0.001$).

6. There were no statistically significant differences between the number of employee users of organisationally owned aircraft based upon user type, $\chi^2(3) = 1.364$, $p = 0.714$.

7. There were statistically significant differences between the number of non-employee users of organisationally owned aircraft based upon user type, $\chi^2(3) = 11.179$, $p = 0.011$. However, none of the pairwise comparisons reached statistical significance.

8. There were no statistically significant differences between the total number of users of organisationally owned aircraft based upon user type, $\chi^2(3) = 4.533$, $p = 0.209$.

9. There were statistically significant differences between the number of users per organisationally owned aircraft based upon user type, $\chi^2(3) = 19.622$, $p < 0.001$. The only statistically significant pairwise comparison showed that the mean rank number of users per organisationally owned aircraft was higher for recreational users (120.09) than for professional users (78.28). No other pairwise comparisons were statistically significant.

Mann–Whitney U tests yielded the following results:

1. Users who had done a course personally owned more aircraft (mean rank = 474.18) than those who had not (mean rank = 419.45), $U = 110465.5$, $z = 3.176$, $p = 0.001$.

2. Users who had passed an OCA personally owned more aircraft (mean rank = 508.20) than those who had not (mean rank = 303.16), $U = 123120$, $z = 11.013$, $p < 0.001$.

3. MFNZ members personally owned more aircraft (mean rank = 503.06) thank non-members (mean rank = 222.55), $U = 105790$, $z = 13.219$, $p < 0.001$.

4. There was not a statistically significant difference in the number of personally owned aircraft between UAVNZ members (mean rank = 416.10) and non-members (mean rank = 449.64), $U = 43554.5$, $z = -1.347$, $p = 0.178$.

5. There was not a statistically significant difference in the number of personally owned aircraft between users who had operated under a Part 102 Operator's Certificate (mean rank = 417.56) and those who had not (mean rank = 448.44), $U = 36388$, $z = -1.130$, $p = 0.258$.

6. There was not a statistically significant difference in the number of other users for personally owned aircraft between users that had attended a course (mean rank = 423.00) and those who had not (mean rank = 407.01), $U = 88657.5$, $z = -1.068$, $p = 0.286$.

7. There was not a statistically significant difference in the number of other users for personally owned aircraft between users that had passed an OCA (mean rank = 415.71) and those who had not (mean rank = 411.58), $U = 71613$, $z = 0.251$, $p = 0.802$.

8. There was not a statistically significant difference in the number of other users for personally owned aircraft between MFNZ members (mean rank = 414.20) and non-members (mean rank = 415.79), $U = 51957$, $z = -0.083$, $p = 0.934$.

9. There was not a statistically significant difference in the number of other users for personally owned aircraft between UAVNZ members (mean rank = 419.96) and non-members (mean rank = 413.66), $U = 40090.5$, $z = 0.286$, $p = 0.775$.

10. There was not a statistically significant difference in the number of other users for personally owned aircraft between those who had operated under a Part 102 Operator's Certificate (mean rank = 440.95) and those who had not (mean rank = 411.43), $U = 34181$, $z = 1.205$, $p = 0.228$.

11. The same results as in points 6–10 were observed for the total number of users for personally owned aircraft (this is because total users is just number of other users plus 1 for the owner of the aircraft).

12. Those who had completed a course had a lower number of users per personally owned aircraft (mean rank = 392.70) than those who had not (mean rank = 433.72), $U = 76903$, $z = -2.466$, $p = 0.014$.

13. Those who had passed an OCA had a lower number of users per personally owned aircraft (mean rank = 361.86) than those who had not (mean rank = 541.97), $U = 40057.5$, $z = -9.869$, $p < 0.001$.

14. MFNZ members had lower numbers of users per personally owned aircraft (mean rank = 367.16) than non-members (mean rank = 620.06), $U = 20295.5$, $z = -11.885$, $p < 0.001$.

15. There was not a statistically significant difference in the number of users per personally owned aircraft between UAVNZ members (mean rank = 439.26) and non-members (mean rank = 410.71), $U = 42213.5$, $z = 1.168$, $p = 0.243$.

16. There was not a statistically significant difference in the number of users per personally owned aircraft between users who had operated under a Part 102 Operator's Certificate (mean rank = 432.46) and those who had not (mean rank = 412.42), $U = 33450.5$, $z = 0.737$, $p = 0.461$.

17. Those who had done a course had higher numbers of organisationally owned aircraft (mean rank = 117.28) than those who had not (mean rank = 86.40), $U = 6593.5$, $z = 3.573$, $p < 0.001$.

18. There was not a statistically significant difference between the number of organisationally owned aircraft based on whether the user had passed an OCA (mean rank = 109.39) or not (mean rank = 96.08), $U = 4297.5$, $z = 1.332$, $p = 0.183$.

19. There was not a statistically significant difference in the number of organisationally owned aircraft between MFNZ members (mean rank = 106.25) and non-members (mean rank = 106.97), $U = 5039$, $z = -0.083$, $p = 0.934$.

20. There was not a statistically significant difference in the number of organisationally owned aircraft between UAVNZ members (mean rank = 118.71) and non-members (mean rank = 103.48), $U = 4083$, $z = 1.474$, $p = 0.141$.

21. Those who had operated under a Part 102 Operator's Certificate had higher numbers of organisationally owned aircraft (mean rank = 130.15) than those who had not (mean rank = 99.58), $U = 5071$, $z = 3.105$, $p = 0.002$.

22. There was not a statistically significant difference in the number of employee users of organisationally owned aircraft between those who had completed a course (mean rank = 101.93) and those who had not (mean rank = 108.95), $U = 5071$, $z = 3.105$, $p = 0.002$.

23. There was not a statistically significant difference in the number of employee users of organisationally owned aircraft between those who had passed an OCA and those who had not, $U = 3312.5$, $z = -1.394$, $p = 0.163$.

24. There was not a statistically significant difference in the number of employee users of organisationally owned aircraft between MFNZ members (mean rank = 104.99) and non-members (mean rank = 109.37), $U = 4864$, $z = -0.501$, $p = 0.616$.

25. UAVNZ members had higher numbers of employee users for organisationally owned aircraft (mean rank = 125.43) than non-members (mean rank = 101.82), $U$ = 4365, $z$ = 2.268, $p$ = 0.023.
26. There was not a statistically significant difference in the number of employee users of organisationally owned aircraft between users who had operated under a Part 102 Operator's Certificate (mean rank = 107.41) and those who had not (mean rank = 106.23), $U$ = 3979.5, $z$ = 0.118, $p$ = 0.906.
27. There was not a statistically significant difference in the number of non-employee users of organisationally owned aircraft between users who had completed a course (mean rank = 103.03) and those who had not (mean rank = 112.98), $U$ = 4626.5, $z$ = −1.418, $p$ = 0.156.
28. There was not a statistically significant difference in the number of non-employee users of organisationally owned aircraft between users who had passed an OCA (mean rank = 107.67) and those who had not (mean rank = 102.28), $U$ = 4012, $z$ = 0.664, $p$ = 0.507.
29. There was not a statistically significant difference in the number of non-employee users of organisationally owned aircraft between MFNZ members (mean rank = 109.91) and non-members (mean rank = 100.01), $U$ = 5547, $z$ = 1.405, $p$ = 0.160.
30. There was not a statistically significant difference in the number of non-employee users of organisationally owned aircraft between UAVNZ members (mean rank = 109.04) and non-members (mean rank = 105.87), $U$ = 3676, $z$ = 0.377, $p$ = 0.706.
31. Those who had operated under a Part 102 Operator's Certificate had lower numbers of non-employee users of organisationally owned aircraft (mean rank = 90.21) than those who had not (mean rank = 111.27), $U$ = 3154, $z$ = −2.634, $p$ = 0.008.
32. There was not a statistically significant difference in the total number of users for organisationally owned aircraft between those who had attended a course (mean rank = 105.50) and those who had not (mean rank = 108.36), $U$ = 4968.5, $z$ = −0.327, $p$ = 0.744.
33. There was not a statistically significant difference in the total number of users for organisationally owned aircraft between those who had passed an OCA (mean rank = 104.77) and those who had not (mean rank = 112.73), $U$ = 3531.5, $z$ = −0.787, $p$ = 0.431.
34. There was not a statistically significant difference in the total number users for organisationally owned aircraft between MFNZ members (mean rank = 108.38) and non-members (mean rank = 102.92), $U$ = 5335, $z$ = 0.623, $p$ = 0.533.
35. UAVNZ members had a higher total number of users for organisationally owned aircraft (mean rank = 125.93) than non-members (mean rank = 101.70), $U$ = 4386, $z$ = 2.318, $p$ = 0.020.
36. Those who had operated under a Part 102 Operator's Certificate had a lower total number of users for organisationally owned aircraft (mean rank = 89.19) than those who had not (mean rank = 111.57), $U$ = 3105, $z$ = −2.248, $p$ = 0.025.
37. Those who had attended a course had a lower number of users per organisationally owned aircraft (mean rank = 95.33) than those who had not (mean rank = 127.34), $U$ = 3464, $z$ = −3.636, $p$ < 0.001.
38. Those who had passed an OCA had a lower number of users per organisationally owned aircraft (mean rank = 101.80) than those who had not (mean rank = 123.46), $U$ = 3038, $z$ = −2.127, $p$ = 0.033.
39. There was not a statistically significant difference in the number of users per organisationally owned aircraft between MFNZ members (mean rank = 106.69) and non-members (mean rank = 106.13), $U$ = 5100.5, $z$ = 0.064, $p$ = 0.949.
40. There was not a statistically significant difference in the number of users per organisationally owned aircraft between UAVNZ members (mean rank = 110.13) and non-members (mean rank = 105.60), $U$ = 3722.5, $z$ = 0.430, $p$ = 0.667.

41. Those who had operated under a Part 102 Operator's Certificate had a lower number of users per organisationally owned aircraft (mean rank = 75.50) than those who had not (mean rank = 115.57), $U = 2448$, $z = -3.996$, $p < 0.001$.

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
