# Peer review of "Reconciling Registration Policies for Unmanned Aircraft with Unmanned Aircraft Ownership Characteristics"

_drones, doi:10.3390/drones7010063_

Round 1

Reviewer 1 Report

-        The manuscript is well-organized but I believe that authors should make some changes. 

-        The section “Conclusions” is not mandatory but should be added to the manuscript because the discussion is long.

-        It would be interesting to include the legislation for drones in New Zealand. 

-        I believe that there is no need for the word “source’ in lines 53,106 and 138.  

-        Please delete the second “in” in the line 82. 

-        The footer from the table 1, I believe that should just be below the table 1 (lanes 111-117). 

Author Response

Thank you for your time spent reviewing this manuscript. Please find attached a document detailing our responses to each comment.

Reviewer 2 Report

The author's research is devoted to a rather urgent problem for the sphere of drones - their registration and accounting. Researchers and practitioners in many countries around the world are working on organizing effective accounting that does not create difficulties for drone users at the same time. In this context, the presented manuscript is interesting and relevant.

As remarks, we can say that the ending is not entirely clear, which scheme the author considers the most optimal, it is not very clear from the text in the "Discussion" section. Perhaps the author should add a section - "Conclusions", and add more specific conclusions in it.

Author Response

(The authors gave the same response as above.)

Round 2

Reviewer 1 Report

I have read the new manuscript and I found that changes have been made. I am satisfied with these changes.